# Clinical Consideration of Anatomical Variations in the Common Hepatic Arteries: An Analysis Using MDCT Angiography

**DOI:** 10.3390/diagnostics13091636

**Published:** 2023-05-05

**Authors:** Laura Andreea Bolintineanu (Ghenciu), Sorin Lucian Bolintineanu, Nicoleta Iacob, Delia-Elena Zăhoi

**Affiliations:** 1Department of Functional Sciences, Victor Babeș University of Medicine and Pharmacy, E. Murgu Sq., No. 2, 300041 Timisoara, Romania; bolintineanu.laura@umft.ro; 2Department of Anatomy and Embryology, Victor Babeș University of Medicine and Pharmacy, E. Murgu Sq., No. 2, 300041 Timisoara, Romania; zahoi.delia@umft.ro; 3Department of Multidetector Computed Tomography and Magnetic Resonance Imaging, Neuromed Diagnostic Imaging Centre, 300218 Timisoara, Romania; nicoiacob@yahoo.co.uk

**Keywords:** hepatic artery, anatomical variations, multidetector computed tomography angiography

## Abstract

Purpose: The purpose of this study was to determine the prevalence of normal hepatic vascularization and variations in the common hepatic arteries using multidetector computer tomography angiography. These variants should be acknowledged before any surgery of the upper abdomen. The aim of our work was to analyze the variations in the hepatic arteries and their possible clinical and surgical implications. Materials and methods: This study was carried out on 4192 patients who underwent 64-slice MDCT angiography, from August 2015 to December 2021. We used surface and volume-rendering techniques in order to post-process images of the vascular components in the desired area. Results: We highlighted 76 cases with replaced common hepatic arteries, which are characterized by the origin of the common hepatic artery trunk located outside the classical composition of the celiac trunk. We identified three levels of origin: the abdominal aorta, the superior mesenteric artery and the left gastric artery. We observed six different aspects of the morphological variability of the celiac trunk and the superior mesenteric artery. The trajectory of the artery trunk, between the aortic origin and the hepatic pedicle portion of the hepatic portal vein, is variable and we analyzed the pancreatic trajectory accordingly. Conclusions: The prevalence of hepatic arterial variants found during this study was similar to that in other specialized studies. We came across variants that have not been described in the well-known classification of Michels and even described extremely rare variations. The study of abnormal hepatic vascularization plays an important role in the surgical planning of hepatic transplantation, liver and pancreatic resection and extrahepatic upper abdominal surgeries.

## 1. Introduction

Clinical implications of the different vascular variations have been studied extensively and reported by many in the medical specialized literature [1,2,3,4]. Knowledge of the anatomical variations in hepatic arteries plays a very important role from the planning to the execution of surgeries involving the liver and pancreas because of the complex vasculature surrounding them. Additionally, it is of great importance in laparoscopic surgeries as well as in interventional radiological procedures, such as bleeding embolization, transcatheter arterial chemoembolization and transarterial radioembolization [1]. Surgical treatment of tumoral lesions involving the pancreas, liver, and duodenum requires highly specialized preoperative imaging in order to observe variants of the vasculature, if present [2]. If a replaced common hepatic artery (RCHA) appears to be present, special precautions are required as extensive damage to the RCHA during supramesocolic abdominal surgery has led to hemorrhages and ischemia [5]. The incidence of vascular complications during complex surgeries is documented within a range varying from 0.7% to 12.9% [6]. Preservation of the replaced common hepatic arteries during surgery is recommended in all cases where it is possible.

A transpancreatic common hepatic artery, which is a variant that appears very rarely in the literature, represents one of the most challenging variations and is present when the RCHA has its pathway throughout the body of the pancreas [7]. It is particularly difficult to save the RCHA during surgical dissection of the pancreas because of its intraparenchymal course.

The celiac trunk (CT) was described by Haller in 1756 for the first time and it represents a source of three arteries for the organs in the supramesocolic part of the abdomen: the left gastric artery (LGA), the splenic artery (SA), and the common hepatic artery (CHA). This morphological and structural aspect, considered standard anatomy, is present with a prevalence that varies between 72.29% [8] and 90.78% [9]. Nowadays, the most common classifications that are being used for the description of the CT and its variants are those described by Morita [10] and Michels [11]. During the planning of surgical interventions involving the abdominal part of the esophagus, stomach, duodenum, pancreas, gallbladder, liver, and spleen, it is of great importance to have knowledge about the possible variants in the vascular pattern of the celiac trunk and its branches [3].

The CHA rises from the CT along with the SA and the LGA [12]. The RCHA represents a rare anatomic variant, which involves a persistent embryonic artery that occurs in the absence of the CHA originating from the CT [1].

MDCT angiography is an accurate radiological investigation that could help surgeons clearly delineate the vascular anatomy presurgery, by identifying different vascular anomalies [13]. It uses faster volume imaging of the entire liver and can deliver thinner slices at high spatial resolution if we compare it with CT imaging with a single detector row [14]. At this stage of the development of imaging examination techniques in the specialized medical research field, MDCT angiography is the best-performing examination method for highlighting significant vascular variations, followed by their storage and archiving [15].

The main purposes of this study were the evaluation of the origin of RCHAs and the anatomic course of these variants using MDCT angiography in a large study. In addition, we analyzed the association between the RCHA and the morphological variations in the CT.

## 2. Materials and Methods

### 2.1. Patient Selection

This study was carried out on 4192 patients who presented themselves for various symptoms at Timis County Emergency Clinical Hospital and were referred to undergo 64-slice MDCT angiography, from August 2015 to December 2021. The patients included in this study did not have any history of personal pathologies of the abdominal organs, nor other vascular pathologies of the supramesocolic compartment of the abdominal cavity. The exclusion criteria we used were: the presence of tumors affecting the upper abdominal compartment, complete occlusion of the CT, any history of gastro–duodenal–pancreatic surgical interventions, history of major allergic reactions to contrast agents that have been used during radiodiagnosis, history of radiation necrosis, severe chronic and/or mental illnesses, uncooperative patients and, lastly, history of claustrophobia. We firstly took into consideration 4315 patients; after we applied these criteria, we analyzed a total of 4192 patients. Patients’ ages were between 18 and 93 years old, belonging to the Romanian population. Alongside a full anamnesis and a complete set of blood investigations, radiological examination was also carried out. This study was conducted according to the guidelines of the Declaration of Helsinki and followed the European Union General Data Protection Regulation (GDPR) [16]. The analysis was approved by the Ethics Committee of the “Victor Babes” University of Medicine and Pharmacy from Timisoara, Romania (protocol code 26/2019).

### 2.2. CT Scanning Protocols

The device that used for the radiological investigation was the 64-slice MDCT system (SOMATOM Sensation, Siemens Medical Solutions, Forchheim, Germany). In the present study, multiplanar reformation (MPR), maximum intensity projection (MIP), volume-rendering techniques (VRT) and shaded surface display (SSD) were used for visualization and in order to post-process images of vascular components in the upper abdominal sector. MPR images were used to visualize the CT. MIPs were created when a specific projection was selected. VRT was used for integration of all information from a volumetric data set by using the images obtained and rendering anatomical structures in different colors. SSD was used mostly to visualize the hepatic parenchyma. The CT examination was performed with a tube voltage of 120 kVp and a reference tube current of 110 mAs. The gantry rotation time was set at 0.33 s and the mean acquisition time was 31.9 s. A detector collimation of 64 × 0.6 mm was used, while the pitch was 0.8. The scanning was performed in a cradiocaudal direction. The images were displayed in axial, coronal and sagittal planes with 5 mm thick slices.

### 2.3. Contrast Material Injection

All patients received iodinated contrast through an 18–20-gauge cannula in an antecubital vein by use of a standard dual-head CT power injector. The volume of the injection was 120 mL at a flow rate of 4 mL/sec. Afterwards, a fixed saline flush was injected at the same flow rate.

### 2.4. Qualitative Image Analysis

Each of the patients included in the present study was monitored during the paraclinical investigations and following that, the images were reconstructed by a team consisting of a radiologist and an anatomist.

### 2.5. Statistical Analysis

We used Microsoft Office Excel 2007 (Washington, DC, USA) for gathering all the cases; for correlating data and determining the distribution, we used IBM SPSS Statistics Version 29 (Chicago, IL, USA). The data processed in Excel were imported into the IBM SPSS program as whole sheets.

All the abbreviations used in this manuscript are listed in Appendix A.

## 3. Results

From the 4192 cases included in the present study, we discovered 800 patients with deviations from the normal supply. Of these 800 cases with variants of the hepatic arteries, we highlighted 76 cases with RCHAs, which are characterized by the origin of the CHA trunk located outside the classical composition of the CT (a “true” CT-hepato–gastro–splenic trunk) or a “false” CT represented by a hepato–splenic trunk, with the LGA originating proximal to the bifurcation level. The prevalence of RCHAs within the entire study was 1.81% (76 cases).

### 3.1. Age and Sex Parameters

Within the group of cases with RCHAs, 71.05% (54 cases out of 76 cases) were male and 28.94% (22 cases out of 76 cases) were female (Table 1). The average age of the group was 66.4 years (with a range varying from 19 to 93 years). For the 54 male cases, the average age was 65 years (with a range between 19 and 86 years). For the 22 female cases, the average age was 72.5 years (with a range between 40 and 93 years).

### 3.2. Level of Origin

Studying the level of origin of RCHAs in the 76 cases, we revealed three distinct morphological types. In their order of frequency, the origin of the RCHA was present at the level of: (i) the abdominal aorta (AA)—69 cases (90.78%); (ii) the superior mesenteric artery (SMA)—6 cases (6.89%); and (iii) the left gastric artery (LGA)—1 case (1.31%) (Figure 1).

In the first group, with RCHAs arising from the AA, we included two subgroups: (i) 20 cases with the absence of the CT as a morphological entity (independent origin of the CHA, the LGA and the SA in the AA) and (ii) 49 cases with the association of a gastro–splenic trunk with the independent origin of the CHA from the AA. We studied the vertebral level of the origin of these 69 cases with RCHAs arising from the AA, and observed that most of the arterial trunks of the CHA originated between the intervertebral disc T12/L1 and the middle 1/3 of the body of the first lumbar vertebra, also with an extension range between the middle 1/3 of the twelfth thoracic vertebra (T12) and the upper 1/3 of the second lumbar vertebra (L2) (69.56%; 48/69 cases). The endoluminal diameter of the RCHA at its origin averaged 0.42 cm, with a range between 0.23 and 0.6 cm.

Discussing the second group, with RCHAs arising from the SMA, we observed that the distance between the origin of RCHA and the origin of the source trunk showed an average of 3.44 cm, with a variation range from 2.34 to 4.51 cm; the endoluminal diameter of RCHA at its origin averaged 0.49 cm, with a range between 0.39 and 0.66 cm.

In the third group, with RCHAs arising from the LGA, the distance between the origin of the RCHA and the origin of the source trunk was 3.31 cm, while the endoluminal diameter of the RCHA at its origin was 0.25 cm.

### 3.3. Aspects of the Morphological Variability of the Celiac Trunk and the Superior Mesenteric Artery Associated with Replaced Common Hepatic Arteries

We highlighted different aspects of the morphological variability of the CT and the SMA in the 76 cases that presented RCHAs (Table 2). We describe six variational aspects, which in descending order of their frequency were:A gastro–splenic trunk associated with the independent origin of the RCHA from the AA in 68.42% of cases (52 cases);CT absent as a morphological entity, with an independent origin of the CHA, the LGA, and the SA from the AA in 26.31% of cases (20 cases);A gastro–splenic–mesenteric trunk with an independent origin of the CHA in the AA in 1.31% of cases (one case);A common trunk formed by left gastric artery–gastroduodenal artery–splenic artery (LGA–GDA–SA), associated with an independent origin of the RCHA from the AA in 1.31% of cases with a RCHA (one case);A gastro–phreno–splenic trunk associated with an independent origin of the SMA from which originated the RCHA, with the presence of a complete inversus site in 1.31% of cases (one case);A gastro–splenic trunk associated with the origin of the RCHA from the LGA in 1.31% of cases (one case) (Figure 2).

### 3.4. Relations of RCHAs with Pancreatic Parenchyma, Hepatic Portal Vein and Superior Mesenteric Vein

The trajectory of the RCHA trunk, between the aortic origin and the hepatic pedicle portion of the hepatic portal vein (HPV), is variable. It depends mostly on the origin of the RCHA and the type of morphological variability associated with it. We observed the trajectory of the RCHA based on these three morphological parameters: (i) relationship with the pancreatic parenchyma; (ii) level of the vascular trajectory according to the pancreatic parenchyma; (iii) relationship with the HPV and the superior mesenteric vein (SMV) path. The vast majority 97.37% (74 cases) had an extrapancreatic trajectory, and only 2.63% (2) had a transpancreatic trajectory (Figure 3).

Within the group of cases with extrapancreatic trajectory, the analysis of the relations of the RCHA trunk with the pancreatic parenchyma highlighted three morphological types: (i) one trajectory superior to the pancreatic parenchyma in 75.68% (56/74 cases); (ii) one trajectory inferior to the pancreas parenchyma in 12.97% (17/74 cases); (iii) one semicircular trajectory inferior to the pancreatic parenchyma in 1.35% (1/74% of cases) in a case of total situs inversus. The discussion of the relations of the RCHA trajectory with the HPV and SMV trajectories led to the description of two subtypes for each of the first two morphological types in the cases of an RCHA with an extrapancreatic pathway. Considering these aspects, within the group of cases with type I, two subtypes can be observed: subtype Ia-RCHA, with a superior trajectory to the pancreas associated with a superior trajectory to the HPV in 1.79% (one case); subtype Ib-RCHA, with a superior trajectory of the pancreas associated with a posterosuperior trajectory to the HPV in 98.21% (55 cases). In the case of type II, two subtypes can be observed: subtype IIa-RCHA, with trajectory posterior to the pancreas associated with a trajectory posterior to the HPV in 94.12% (16 cases); subtype Iib-RCHA, with a trajectory posterior to the pancreas associated with a trajectory posterior to the HPV and the SMV in 5.88% (one case).

Within the group of cases of an RCHA with a transpancreatic trajectory (type IV), the analysis of the relation of the RCHA trunk with the pancreatic parenchyma revealed two morphological subtypes of relationships according to the existing relations with: (i) the confluence of the SMV with the HPV (subtype Iva); (ii) the SMV trunk (subtype Ivb). Discussing subtype Iva (50% of cases), the RCHA is placed anteriorly to the confluence between the SMV and the HPV. Meanwhile, in the subtype Ivb (50% of cases), the RCHA is placed anteriorly to the trunk of the HPV.

## 4. Discussion

CT variants and their associated pathologies are commonly found in the general population. Using new innovative technologies, such as MDCT angiography, one can perform a preoperative, non-invasive reconstruction of the status of the individual vascularization in each patient [17]. In the specialized literature, data vary across radiological and cadaveric studies. In the present study, we applied MDCT angiography in the diagnosis of the replaced common hepatic arteries.

Zimmitti et al. [18] emphasize the presence of RCHAs in the specialized literature, with a variation range between 0.4 and 4.5%. Winston observes variations in the origin of the common hepatic artery in 4% of the cases studied in his paper, of which 2% showed the origin of the CHA in the SMA and 2% showed the origin of the CHA in the AA [19]. An RCHA arising from the SMA was reportedly seen in 1.13% of 19,013 cases in a systematic review of the literature. [4]

Prakash et al. [3] analyzed the discussions on the variations in the CT from the specialized literature. His study using a different method (dissection) presented the existence of a percentage of 86% of CT cases according to the standard. The other 14% of cases were divided into three categories: (1) the LGA, the CHA and the SA originated separately from the AA (4%); (2) the LGA arose from the AA, and the CT was bifurcated, giving rise to the CHA and the SA (4.8%); (3) the CT bifurcated into the CHA and the LGA, while the SA arose from the AA.

Song et al. [20] studied the prevalence of CT variations on 5002 subjects. With the exclusion of the ambiguous anatomy of the CT, 4756 patients (96.3%) presented the anatomy of the CHA according to the standard, with its origin in the CT. The other 183 patients were divided according to the origin of the RCHA: (1) RCHAs arose from the LGA in 0.16% of cases (2) RCHAs arose from the SMA in 3% of cases, and (3) RCHAs arose from the AA in 0.4% of cases. Our study shows three levels of origin for the RCHA, the most common level of origin of RCHAs being the AA, with a percentage of 1.64% of all cases studied. This level of origin is higher than others found in the literature. In the their order of frequency, the next level of origin is the SMA, with a prevalence of 0.14% of all cases studied. The last level of origin of RCHAs is the LGA, with one case out of all cases studied, therefore a prevalence of 0.002%.

Thangarajah et al. [21], in an analysis of anatomical variations in the CT, performed on 200 cases, observed a standard CT configuration in 89.5% of cases studied, while the rest of the patients showed CT variations or ambiguous anatomy of the CT (not included in any of the criteria used by Michels). A total of 1.5% of the total cases presented RCHAs with origin in the SMA, 0.5% showed RCHAs with origin in the LGA, and none presented RCHAs with independent origin from the AA. The CHA, as a branch from the AA, is one of the rarest variants reported in the literature. Cankal et al. reported RCHAs arising from the AA as the most common variant (2%), with the next variant in order of frequency being RCHAs from the SMA (1.5%) and no case with RCHAs from the LGA was reported [22].

According to Hiatt [23], out of the 1000 cases studied, 1.5% presented RCHAs with an origin at the SMA level, while 0.2% of cases had RCHAs with origin in the abdominal aorta. Malvyva also highlighted, from a total of 110 cases studied, 2 cases of RCHAs branching from the SMA (1.82%), while no case with RCHAs with the origin in the AA was found [24]. A total of 2% of cases with RCHAs with an origin from the SMA and 1% of cases with RCHAs with an origin from the AA were reported in the study of Ugurel, observing 100 patients [25].

The pathway of the RCHA from its origin to the hepatic pedicular portion of the HPV may be variable and it depends on the level of the artery origin. In situations when replaced hepatic arteries are observed, it is of great importance to acknowledge the anatomy for the complete preservation of vascularization, as well as for an optimal surgical and oncological result [26]. This also involves the study of the peripancreatic pathway of the RCHA trunk before pancreaticoduodenal surgery.

In his study, Song et al. [20] showed that the vast majority of RCHAs presented an extrapancreatic pathway (93.75%). Thus, in the case of patients with RCHAs with origin in the LGA, all showed an extrapancreatic pathway, just like in patients with origin in the AA. In patients with RCHAs originating in the SMA, 11 presented an intrapancreatic pathway.

Ha et al. [13] studied RCHAs with their origin at the SMA level and their pancreatic relations, defining the following morphological subtypes: (1) the RCHA crosses the pancreas (transpancreatic pathway), posterior to the superior mesenteric vein, (2) the RCHA does not cross the pancreas (extrapancreatic pathway) and is situated posterior to the trunk of the HPV/SMV and (3) the RCHA does not cross the pancreas (extrapancreatic pathway) and is situated anterior to the HPV or is situated posterior to the SMV.

Yang et al. analyzed 1324 cases using computer tomography and angiography, observing that 2.34% of them presented replaced common hepatic arteries originating in the SMA [27].

Noussios et al. performed a systematic database search of the scientific literature and found that the prevalence of RCHAs arising from the SMA by a common trunk that is referred to as the hepatomesenteric trunk is 1.2%, while the incidence in the literature for this variant ranges from 0.4% to 4.5% [28].

The presence of an RCHA crossing the pancreatic parenchyma is quite rare. Still, recognition of this variant is known to have important clinical and surgical implications in multiorgan transplantation and pancreatic surgery. If present, transpancreatic replaced common hepatic arteries may lead to intraoperative hemorrhage or may even compromise hepatic arterial inflow. Therefore, the radiologist and the surgeon should both analyze preoperative imaging in order to trace abnormal anatomy of the celiac trunk, the superior mesenteric artery, and their branches [29].

Ishigami et al. studied 788 liver transplant donor candidates who had undergone MDCT angiography in order to assess standard or ambiguous vascular anatomy [7]. They observed tp-CHA variation in 0.38% of cases. However, one recent case series assessed nine patients undergoing pancreaticoduodenectomy with an RCHA and found that the intrahepatic course occurred more often than it appears in the literature, with a prevalence of 33% [30].

One of the explanations for the unequal gender distribution (65 vs. 72.5) of the hepatic arterial variants could be that many diseases preferentially affect men, also from a younger age (cardiovascular, hepatobiliary or malignant diseases) [6].

As a study with a large study group, this paper highlighted many rare anatomic variations that have not been categorized by the Michels and Hiatt classification systems. Although there have been previous studies reporting RCHAs originating from the LGA [20,21], the prevalence rates in our study were 1.31%, higher than those mentioned in other studies. A tp-CHA is another extremely rare anatomic variant identified in this study. Finally, to the best of our knowledge, there have been no previous studies reporting a RCHA with its origin from the SMA associated with a gastro–phreno–splenic trunk.

Compared to a few studies that used MDCT angiography as the paraclinical investigation, some cadaveric studies show higher rates of these variations in hepatic arteries because the usage of contrast agents may lead to insufficiency of opacification in extremely thin vessels [22]. One other explanation could be that cadaveric studies use smaller sample sizes. Still, in some studies, it was also argued that radiological studies can better detect variations [31]. Sukumaran et al. observed RCHAs from the SMA in 2.86% of cases in a cadaveric study on 35 specimens [32], while Ghosh highlighted the same variation in 1.6% cases in a study on 125 cadavers [33].

Regarding the limitations of this present study, we can discuss the fact that these variations in the hepatic arteries are only seen from anatomist and radiologist points of view, so a surgeon’s point of view, using cadaveric studies, would be useful. These types of studies, although more accurate, are more difficult to obtain because high numbers of cadavers are needed. Further, given the few studies regarding replaced common hepatic arteries from the specialized literature, larger populations of patients should be evaluated in order to thoroughly define variations in hepatic arteries.

## 5. Conclusions

Variations in hepatic arteries among Romanians have a slightly different distribution when compared to such variations in other countries. On the basis of the review of the MDCT angiography images of 4192 patients, we provided a comprehensive list of common variations in hepatic arteries. Surgeons and radiologists should acknowledge the replaced common origin and course of hepatic arteries and before any surgical interventions, as it this reduce complications and improve patient outcomes.

## Figures and Tables

**Figure 1 diagnostics-13-01636-f001:**
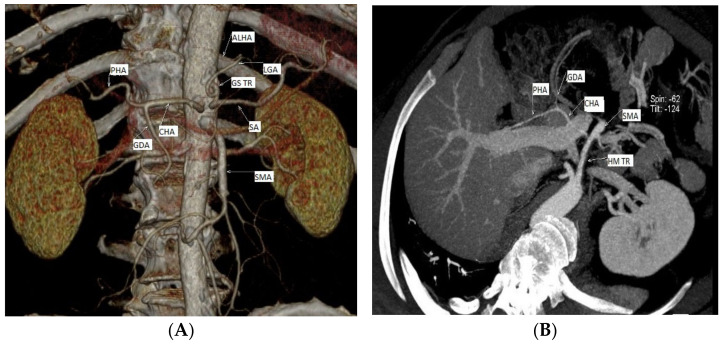
Three distinct types of level of origin of the replaced common hepatic arteries. Replaced CHA origins at the level of: (**A**) the abdominal portion of the aorta; (**B**,**C**) the superior mesenteric artery; (**D**) the left gastric artery (personal casuistry). (**A**) Male patient, 75 years old, with a diagnosis of peripheral arterial disease (PAD); (**B**) male patient, 66 years old, with a diagnosis of critical limb ischemia; 3D reconstruction, for a better view of the described arteries we used gantry tilt/spin (degrees -124/42) (**C**) male patient, 70 years old, with a diagnosis of chronic kidney disease; (**D**) male patient, 73 years old, with a diagnosis of PAD. Abbreviations: ALHA—accessory left hepatic artery, CHA—common hepatic artery, CT—celiac trunk, GS TR—gastrosplenic trunk GDA—gastroduodenal artery, HM TR—hepatomesenteric trunk LGA—left gastric artery, LHA—left hepatic artery, PHA—proper hepatic artery, SMA—superior mesenteric artery, and SA—splenic artery.

**Figure 2 diagnostics-13-01636-f002:**
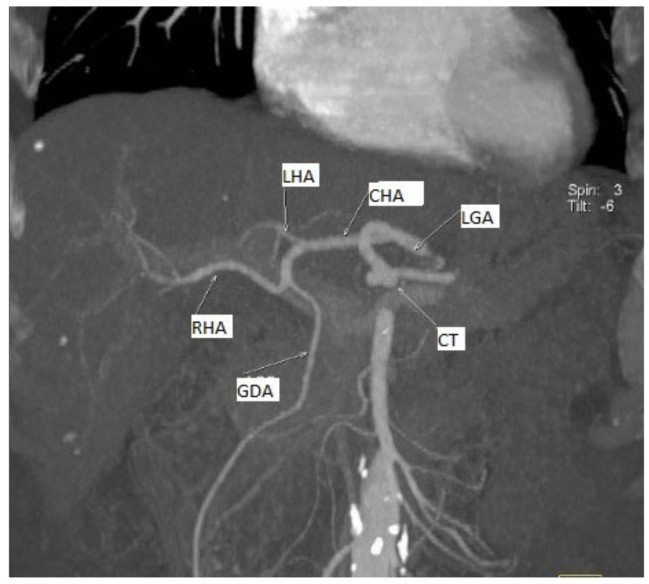
MDCT angiography, coronal view. 3D reconstruction, for a better view of the described arteries we used gantry tilt/spin (degrees -6/3). Replaced common hepatic artery originating from the left gastric artery. Female patient, 66 years old, with a diagnosis of PAD.

**Figure 3 diagnostics-13-01636-f003:**
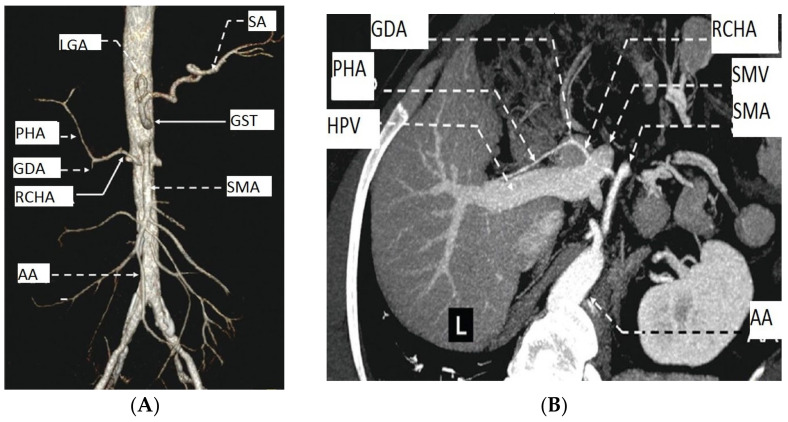
MDCT angiography of the AA and liver. Female patient, 83 years old, with a diagnosis of PAD. (**A**) Anterior aspect; (**B**) transversal aspect. Abbreviations: HPV—hepatic portal vein, PHA—proper hepatic artery, SMV—superior mesenteric vein, and L—liver.

**Table 1 diagnostics-13-01636-t001:** Age, sex and condition distribution.

Variable	Female	%	Male	%	Total	%
Age group	22	28.94	54	72.05	76	100
11–20	1	1.31	1	1.31	2	2.63
21–40	4	5.26	7	9.21	11	14.47
41–60	8	10.52	16	21.05	24	31.57
61–80	8	10.52	26	34,21	34	44.73
over 80	1	1.31	4	5.26	5	6.57
Disease						
cardiovascular	18	23.68	42	55.26	60	78.94
respiratory	-		2	2.63	2	2.63
renal	2	2.63	2	2.63	4	5.26
other	2	2.63	8	10.52	10	13.15

**Table 2 diagnostics-13-01636-t002:** Morphological variations in the celiac trunk and the course of replaced common hepatic arteries.

		TOTAL	GSM Trunk	Absent CT	Gastro–Splenic Trunk	LGA–GDA–SA Trunk	Situs Inversus
A. EXTRAPANCREATIC							
I. Superior to pancreas							
Ia	Superior to HPV	1	-	-	1	-	-
Ib	Postero-superior to HPV	55	-	18	37	-	-
II. Posterior to pancreas							
IIa	Posterior to HPV	16	-	2	14	-	-
IIb	Posterior to HPV and SMV	1	-	-	-	1	-
III. Inferior semicircular							
IIIa	Extrapancreatic	1	-	-	-	-	1
B. TRANSPANCREATIC							
IV. Transpancreatic							
IVa	Anterior to confluent HPV and SMV	1	-	-	1	-	-
IVb	Anterior to SMV	1	1	-	-	-	-
	TOTAL	76	1	20	53	1	1

Abbreviations: CT—celiac trunk; RCHA—replaced common hepatic artery; GSM trunk—gastro—spleno—mesenteric trunk; LGA—GDA—SA trunk—left gastric artery—gastroduodenal artery—splenic artery trunk; HPV—hepatic portal vein; SMV—superior mesenteric vein.

## Data Availability

The data presented in this study are available on request from the corresponding author. The data are not publicly available due to ongoing unpublished research.

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
