# Peer review of "Clinical Consideration of Anatomical Variations in the Common Hepatic Arteries: An Analysis Using MDCT Angiography"

_diagnostics, 2023, doi:10.3390/diagnostics13091636_

Round 1

Reviewer 1 Report

The article describes a MDCT angiography study to determine anatomical variants of the common hepatic arteries. The article falls within the scope of diagnostics. The article is interesting.

The revision improved the manuscript.

There is a moderate number of typographic errors:

Line 151: replace: “We can easily observe” by “We observed”.

Line 171 and line 250: replace: “Disscusing” by “Discussing”.

Line 301: replace: “observind” by “observing”.

Line 353 and 354: delete all words (triple repetition).

Line 369: replace: “sample studies” by “sample sizes”.

Line 371: replace: “cases in cadaveric studies” by “of cases in a cadaveric study”.

Line 370: replace: “varations” by “variations”.

Line 374: replace: “disscus” by “discuss”.

Line 377: replace: “altough” by “although”.

Line 377-378: replace: “large lots” by “high numbers”.

Line 379: replace: “pacients” by “patients”.

Line 383-385: delete lines.

Line 387: replace: “assesed” by “assessed”.

Line 391: replace: “being” by “representing”.

Author Response

Dear reviewer,

Firstly, I would like to express my sincere apologies for the manuscript you were provided with. We have made last minutes changes to the manuscript and didn't mean to share these changes, but although we stopped the 'track changes' button, I have been told that the changes still appeared and maybe you had a more difficult time reading the manuscript. I am so sorry about that. 

I appreciate greatly the precise and valuable observations. I am glad that you find this topic interesting.  I made all the changes you suggested, the minor spell checks and deleted the sentences which appeared to say the same information elsewhere in the text 

Once again, we thank you for your time.

With utmost respect,

Assist.prof. Ghenciu Laura Andreea

Reviewer 2 Report

Article on the anatomical variability of the celiac trunk 

Well written article but you don't understand what the new elements are compared to previous studies 

Not fluent English

The main topic of the article is not understood

Author Response

Dear reviewer,

Firstly, I would like to express my sincere apologies for the manuscript you were provided with. We have made last minutes changes to the manuscript and didn't mean to share these changes, but although we stopped the 'track changes' button, I have been told that the changes still appeared and maybe you had a more difficult time reading the manuscript. I am so sorry about that. 

Even though the variation of replaced common hepatic arteries is well known, it is often overlooked by radiologists. We have tried to emphasize its importance, as well as the clinical relevance by including in this study a large number of cases, as knowledge of its presence might reduce complications and also improve postsurgical outcomes. There aren't many studies across the literature with large study lots, especially regarding the pathway of RCHA and many articles published are case studies.  A transpancreatic RCHA is very rare and has been reported in our study. Also, the origin of RCHAs in our study was slightly different than the ones reported before. This is why we think that our study is interesting, even though it is not a new topic and variations of the hepatic arteries have been studied before.

I have made spell check, deleted a few sentences that repeated the information, added a few more pieces of information about the MDCT and changed the conclusion part into a more readable and understandable version.

Once again, we thank you for your time.

With utmost respect,

Assist.prof. Ghenciu Laura Andreea

Reviewer 3 Report

This interesting work by Bolintineanu et al. is focused on the prevalence of normal hepatic vascularization and variations of the common hepatic arteries using Multidetector Computer Tomography Angiography on 4192 patients. It plays an important role in the surgical planning of hepatic transplantation, liver and pancreatic resection and extrahepatic upper abdominal surgeries.

I have few questions:

1) The inclusion criteria are not very well explained in the text: which type of patient was included in the study? Were they all patients with vascular deseases or did they belong to the general population? In my opinion, this could also change the finding of such anatomical anomalies. I would propose to provide a summary table of the characteristics of the population with a clear indication to the CT scan.

2) Also, was a written informed consent signed? I do not understand the claustrophobic exclusion criterion to perform a CT scan, have you observed many drop out for this reason? Was it considered a feasibility criteria of the exam?

3) there are several repetitions in the draft that I could visualize:

Page 3, line 100

Page 4, line 143

page 11, lines 353-354-355

page 11, lines 383-384 -385

4) Figure 1 is not clear at all, the writings are too small to be viewable and furthermore the captions appear all erased. It should be revised.

5) there are several lexical errors (eg: page 1, line 19: pacients (patients?); page 6, line171: Discussing (?)). English should be revised

Author Response

Dear reviewer,

Firstly, I would like to express my sincere apologies for the manuscript you were provided with. We have made last minutes changes to the manuscript and didn't mean to share these changes, but although we stopped the 'track changes' button, I have been told that the changes still appeared and maybe you had a more difficult time reading the manuscript. I am so sorry about that. 

I greatly appreciate the precise and valuable observations and made all the changes that you suggested. I will address all of them separately in the next reply:

  1. The inclusion criteria are not very well explained in the text: which type of patient was included in the study? Were they all patients with vascular deseases or did they belong to the general population? In my opinion, this could also change the finding of such anatomical anomalies. I would propose to provide a summary table of the characteristics of the population with a clear indication to the CT scan.

          Most of the patients had a vascular disease (most of them critical ischemia or peripheral arterial disease), while a few had other diseases. They were examined at the hospital depending on their symptomatology and were referred to undergo MDCT. I also added a table of the patients with age, gender and disease. 

2. Also, was a written informed consent signed? I do not understand the claustrophobic exclusion criterion to perform a CT scan, have you observed many drop out for this reason? Was it considered a feasibility criteria of the exam?

yes, we stated in the footnotes that informed consent was obtained from all subjects involved in the study. Thank you for your question about claustrophobia. in our radiologists' experience there have been a few cases with severe claustrophobia. which is why we included this criteria. I should mention that only severe claustrophobia with personal history was taken into consideration, as it could lead to stopping the MDCT during the investigation and also lead to artifacts and therefore exclude the case

3. I erased the repetitions from the manuscript, thank you very much for the observations

4. we enlarged the three different pictures and also added one more with volume rendering technique. I hope that now it is more clear.

5. I made spell checks in the entire manuscript. 

Once again, we thank you for your time.

With utmost respect,

Assist.prof. Ghenciu Laura Andreea

Reviewer 4 Report

This study aimed to investigate the prevalence of normal hepatic vascularization and variations of common hepatic arteries using MDCT angiography. However, there are several concerns with this work, as follows:

·         The manuscript contains a lot of marking and changes, making it difficult to read. The authors should provide a clean copy of the manuscript without any markings and corrections.

·         Abstract, Background: The subsection titled Background should provide the background of the study rather than the aim and purpose. If the authors wish to mention the purpose of the study, they should change the subsection to Purpose.

·         Abstract, Materials and methods: The Materials and methods subsection should be more detailed than just one sentence.

·         Introduction, L36-37: The authors should provide references to support the studies mentioned in the literature review.

·         Introduction, L51-53: The authors should explain why the trans-pancreatic common hepatic artery represents one of the most challenging variations in the study.

·         Introduction, L67-70: The authors should provide more background information about MDCT angiography, as it is related to this study.

·         Material and Method, L78-80: The authors should state the time period during which the cases were acquired.

·         Material and Method, L81: The authors should use "did not" instead of "didn't."

·         Section 2.1: It would be helpful to have a table showing patient data with gender, age, and disease.

·         Section 2.5:, L127-128: The authors should clarify whether they mean "IBM" or "IMB" in their writing.

·         The authors should update the version of IBM SPSS mentioned in the manuscript, as the current version is now 29.

·         Table 1 of the Abbreviations can be relocated to the Appendix section.

·         Conclusion, L383-386: The conclusion contains four lines of repeated sentences, which should be revised.

Author Response

Dear reviewer,

Firstly, I would like to express my sincere apologies for the manuscript you were provided with. We have made last minutes changes to the manuscript and didn't mean to share these changes, but although we stopped the 'track changes' button, I have been told that the changes still appeared and maybe you had a more difficult time reading the manuscript. I am so sorry about that. 

I greatly appreciate the precise and valuable observations and made all the changes that you suggested. I will address all of them separately in the next reply:

 1. Abstract, Background: The subsection titled Background should provide the background of the study rather than the aim and purpose. If the authors wish to mention the purpose of the study, they should change the subsection to Purpose., Materials and methods: The Materials and methods subsection should be more detailed than just one sentence.

I changed the paragraph to Purpose and also added a sentence about the MDCT in the abstract. I tried to keep it concise in order not to exceed the number of words in the abstract. Thank you very much. 

2. Introduction, L36-37: The authors should provide references to support the studies mentioned in the literature review,  L51-53: The authors should explain why the trans-pancreatic common hepatic artery represents one of the most challenging variations in the study, L67-70: The authors should provide more background information about MDCT angiography, as it is related to this study.

We provided the references that stated the clinical implication of RCHAs, I explained why a transpancreatic is a challenge during surgery and also added a few more information about the MDCT angiography

3. Material and Method, L78-80: The authors should state the time period during which the cases were acquired, L81: The authors should use "did not" instead of "didn't.". 

I stated the time period (which is August 2015-December 2021) and also made the change.

4. Section 2.1: It would be helpful to have a table showing patient data with gender, age, and disease.

I added a table with the patients containing these 3 variables, age, gender and type of disease

5. L127-128: The authors should clarify whether they mean "IBM" or "IMB" in their writing, Table 1 of the Abbreviations can be relocated to the Appendix section.

I moved the table to the Appendix section and also changed the version and the misspelled word. thank you very much

6. I changed the conclusion into, I hope, a clearer and more understandable version. 

Once again, we thank you for your time.

With utmost respect,

Assist.prof. Ghenciu Laura Andreea

Round 2

Reviewer 2 Report

some improvements have been made

There's still a little work to do...it's starting to be more than acceptable work

Reviewer 4 Report

The authors have addressed all my concerns with corresponding modifications in the revision. I am satisfied with their responses.